# Disruption of Zika Virus xrRNA1-Dependent sfRNA1 Production Results in Tissue-Specific Attenuated Viral Replication

**DOI:** 10.3390/v12101177

**Published:** 2020-10-18

**Authors:** Hadrian Sparks, Brendan Monogue, Benjamin Akiyama, Jeffrey Kieft, J. David Beckham

**Affiliations:** 1Department of Immunology & Microbiology, University of Colorado Anschutz Medical Campus, Aurora, CO 80045, USA; hadrian.sparks@cuanschutz.edu (H.S.); brendan.monogue@cuanschutz.edu (B.M.); 2Department of Biochemistry and Molecular Genetics and 4RNA BioScience Initiative, University of Colorado Denver School of Medicine, Aurora, CO 80045, USA; Benjamin.akiyama@cuanschutz.edu (B.A.); jeffrey.kieft@cuanschutz.edu (J.K.); 3Department of Medicine, Division of Infectious Diseases, University of Colorado School of Medicine, Aurora, CO 80045, USA; 4Rocky Mountain Regional VA Medical Center, Aurora, CO 80045, USA

**Keywords:** Zika virus, flavivirus, RNA structure, non-coding RNA, replication, antibody

## Abstract

The Zika virus (ZIKV), like other flaviviruses, produces several species of sub-genomic RNAs (sfRNAs) during infection, corresponding to noncoding RNA fragments of different lengths that result from the exonuclease degradation of the viral 3′ untranslated region (UTR). Over the course of infection, these sfRNAs accumulate in the cell as a result of an incomplete viral genome degradation of the 3′ UTR by the host 5′ to 3′ exoribonuclease, Xrn1. The halting of Xrn1 in the 3′ UTR is due to two RNA pseudoknot structures in the 3′ UTR, termed exoribonuclease-resistant RNA1 and 2 (xrRNA1&2). Studies with related flaviviruses have shown that sfRNAs are important for pathogenicity and inhibiting both mosquito and mammalian host defense mechanisms. However, these investigations have not included ZIKV and there is very limited data addressing how sfRNAs impact infection in a whole animal model or specific tissues. In this study, we generate a sfRNA1-deficient ZIKV (X1) by targeted mutation in the xrRNA1 3′ UTR structure. We find that the X1 virus lacks the production of the largest ZIKV sfRNA species, sfRNA1. Using the X1 virus to infect adult *Ifnar1*^−*/*−^ mice, we find that while the lack of sfRNA1 does not alter ZIKV replication in the spleen, there is a significant reduction of ZIKV genome replication in the brain and placenta compared to wild-type ZIKV infection. Despite the attenuated phenotype of the X1 ZIKV, mice develop a robust neutralizing antibody response. We conclude that the targeted disruption of xrRNA1 results in tissue-specific attenuation while still supporting robust neutralizing antibody responses. Future studies will need to investigate the tissue-specific mechanisms by which ZIKV sfRNAs influence infection and may utilize targeted xrRNA mutations to develop novel attenuated flavivirus vaccine approaches.

## 1. Introduction

Arthropod-borne flaviviruses such as West Nile virus (WNV), dengue virus (DENV), and Japanese encephalitis virus (JEV) are important human pathogens. More recently, the Zika virus (ZIKV) pandemic and continued increase of tick-borne encephalitis virus infections emphasize the ongoing need to better define the mechanisms of flavivirus pathogenesis. The flavivirus genome is a positive-sense, single-stranded RNA encoding a 5′ untranslated region (UTR), a single open reading frame (ORF), and a highly structured and conserved 3′ UTR. One of these conserved RNA structures in the 3′ UTR is known to inhibit the degradation of viral RNA by host exoribonucleases [1]. These exoribonuclease-resistant RNAs (xrRNAs) form a pseudoknot structure dependent on a complex tertiary structure which protects the remaining 3′ UTR and results in the accumulation of sub-genomic flaviviral RNAs (sfRNAs) during infection [1,2,3]. One or more xrRNAs and the biogenesis of sfRNAs has been identified for several flaviviruses including WNV, Murray Valley encephalitis virus (MVE), JEV, DENV, and ZIKV [1,2,3,4,5].

The specific role of sfRNAs during infection is not well-defined, especially *in vivo*. Recent *in vitro* studies have indicated that sfRNAs are important for the limitation of cytoplasmic mRNA decay, the induction of cell apoptosis, the control of the mammalian host type I interferon (IFN) response, host switching, and the inhibition of RNAi and Toll receptor pathways in arthropod hosts [2,3,4,5,6,7,8,9,10,11]. Due to the lack of structural data for flavivirus xrRNAs, many studies have relied on extensive modifications in the 3′ UTR to completely eliminate the biogenesis of all sfRNA species. With the recent publication of detailed structural data for the first of two xrRNAs in ZIKV (xrRNA1), targeted mutations can now be utilized to disrupt RNA tertiary structure independent of major sequence changes in the 3′ UTR [1]. Using this approach, we made specific mutations in xrRNA1 that disrupt the tertiary structure without significantly altering the 3′ UTR sequence.

Our approach allows us to evaluate the function of individual sfRNAs while maintaining the sequence of sfRNAs. This strategy limits the potential for off-target effects that are due to significant sequence or structural alteration of the 3′ UTR. Both WNV and DENV produce multiple sfRNA species, largely in correlation to the number of xrRNAs encoded in the 3′ UTR. Similarly, ZIKV generates two consistently observed species of sfRNAs (sfRNA1 and sfRNA2), due to the presence of two xrRNA structures (xrRNA1 and xrRNA2) within the 3′ UTR. Despite this, only limited data has been presented that investigates the role of individual sfRNA species [2,3,4,5,6,7,8,9,10,11,12]. Perhaps more importantly, the role of sfRNA production during ZIKV infection in the mouse model is not known.

Here, we use our previous structural data of ZIKV xRNA1 tertiary structure to generate an infectious ZIKV clone in which a discrete, single nucleotide mutation eliminates the production of sfRNA1 without significant changes to the 3′ UTR sequence or structure. We show that the loss of sfRNA1 does not impact vial growth in mammalian cells nor limits infection in mosquito cells. In type I interferon receptor knockout mice, (*Ifnar1*^−*/*−^), ZIKV clones with the xrRNA1 mutation exhibit significantly decreased viral growth in the brain and placenta while still producing a robust neutralizing antibody response. These data show that sfRNA1 production plays a tissue-specific role in the support of viral replication.

## 2. Materials and Methods

### 2.1. Cell Lines and Viruses

African green monkey kidney epithelial cells (Vero E6), human lung epithelial cells (A549), and Aedes albopictus cells (C6/36) were sourced from the American Type Culture Collection (ATCC). Vero and C6/36 cells were cultured in Eagle’s minimum essential medium (MEM), A549 cells were cultured in Ham’s F-12K medium. Both MEM and Ham’s F-12K medium were supplemented with 1 mM sodium pyruvate, 1X non-essential amino acids (100X; ThermoFisher scientific), 100 U/mL streptomycin, 100 µg/mL streptomycin, 10 mM HEPES, and 10% fetal bovine serum (FBS). Dcr-2-expressing Aedes albopictus cells (U4.4) were generously provided by Dr. Aaron Brault (CDC, Division of Vector-Borne Diseases) and cultured in Mitsuhashi and Maramorosch insect medium (M&M) supplemented with 10% FBS, 1X non-essential amino acids, 100 U/mL streptomycin, and 100 µg/mL streptomycin. Mammalian-derived cells were maintained at 37 °C with 5% CO_2_ while Aedes-derived cells were maintained at 28 °C with 5% CO_2_. Viruses used in this study include ZIKV Puerto Rico isolate PRVABC59, a wild-type (WT) clone derived from PRVABC59, and a clone-derived ZIKV with a targeted mutation in xrRNA1 termed X1 virus [13].

### 2.2. Plasmids and Generation of the X1 Mutant

Previously described pACYC177 vector plasmids containing ZIKV PRVABC59 genome from either the 5′ UTR to nt 3498 (p1-ZIKV) or from nt 3109 to the end of the 3′ UTR (p2-ZIKV) were used to generate WT ZIKV or the X1 mutant. Primers ZIKV X1 C35G F (5′-TCCCCAAGCTGTGCCTGACTAGCAGGC-3′) and ZIKV X1 C35G R (5′-GCCTGCTAGTCAGGCACAGCTTGGGGA-3′) were used with a QuikChange II XL Site-Directed Mutagenesis Kit (Agilent; Santa Clara, CA) to introduce the X1 C10415G mutation into the p2-ZIKV plasmid. The resulting X1 mutant p2-ZIKV, untreated WT p2-ZIKV, and p1-ZIKV plasmids were rescued and amplified via rolling circle amplification as described [13]. Prior to in vitro transcription of the viral genomes, the presence of the C10415G mutation in the X1 mutant was confirmed via Sanger sequencing (Eton Bioscience: San Diego, CA, USA).

### 2.3. Rescue and Propagation of ZIKV

The viral RNA genome for both WT ZIKV and the X1 mutant was in vitro transcribed from ligated plasmid DNA using a HiScribe T^7^ ARCA mRNA kit (NEB). Approximately 40 µg of the resulting mRNAs were transfected into Vero cells using MessengerMAX lipofectamine transfection reagent (Invitrogen). Supernatant was collected from transfected cells once 50%–60% cell clearance was observed and spun down to eliminate cell debris. Clarified supernatant containing the rescued WT or X1 virus was aliquoted stored at −80 °C for later use. Extracellular viral RNA and the infectious virus were quantified as detailed below to evaluate the success of the viral rescue. Both rescued WT and X1 viruses were subsequently passaged once in C6/36 cells to increase viral titer.

### 2.4. Virus Quantification

The amount of the cell-free infectious virus in the supernatant of infected cells was quantified using a focus-forming unit assay, as previously described [14]. Mouse anti-Flavivirus group antigen antibody, clone D1-4G2-4-15 (Millipore) was used as primary antibody and donkey anti-mouse IgG antibody conjugated with horseradish peroxidase (Jackson Research Laboratories) served as the secondary antibody. Extracellular viral RNA was extracted from the supernatant of infected cells or serum from infected animals using an E.Z.N.A. viral RNA kit (Omega Bio-Tek) and used to quantify the viral genome present via the following RT-qPCR protocol. Extracted RNA was reverse transcribed using iScript cDNA synthesis (Bio-Rad). Primers Zika1087 (5′-CCGCTGCCCAACACAAG-3′), Zika1163c (5′-CCACTAACGTTCTTTTGCAGACAT-3′), and FAM-tagged probe Zika1108FAM (5′-AGCCTACCTTGACAAGCAGTCAGACACTCAA-3′) were used in combination with a standard curve spanning 107 copies/reaction to 1 copy/reaction to quantify ZIKV genome copies via qPCR. Transformed data were presented as (log10) viral genome copies per microliter.

### 2.5. Northern Blot

Total RNA was isolated from A549 cells infected with either WT or X1 ZIKV and used to evaluate the presence of sfRNAs via Northern blot, as previously described [1].

### 2.6. In Vitro Viral Growth Kinetics Comparison

In 24-well plates, 2 × 10^4^ A549 cells or 4 × 10^4^ U4.4 cells per well were seeded. Cells were infected with either WT or X1 virus at a multiplicity of infection (MOI) of 0.1 for 1 h then washed with 1XPBS to remove extracellular virus before adding back typical cell growth media. At 0, 24, 48, and 72 h post infection (HPI), supernatant from infected cells was collected and used to quantify the amount of infectious virus [via focus-forming unit (FFU) assay] or viral genome (via RT-qPCR) present.

### 2.7. Animal Studies

All animal and infectious disease studies were reviewed by the University of Colorado Institutional Animal Care and Use Committee (IACUC protocol #0047, 6/1/2018) and Institutional Biosafety Committee (#1098, 6/15/2018). C57BL/6 *Ifnar1*^−*/*−^ mice purchased from Jackson Laboratories and C57BL/6 mice expressing human STAT2 protein (h*STAT2* KI) graciously provided by Dr. Michael Diamond (Washington University School of Medicine) were maintained and bred in specific-pathogen-free facilities at the University of Colorado Anschutz Medical Campus animal facility. Animals to be infected were housed in an animal BSL-3 (ABSL-3) laboratory. After infection, mice were observed daily for signs of disease, weight loss, or other terminal indicators until the experiment endpoint.

### 2.8. Characterization of Infection in Adult Ifnar1^−/−^ Mice

Similar numbers of male and female 5–7-week-old *Ifnar1*^−*/*−^ mice were intraperitoneally infected with 1 × 10^4^ FFU of either X1, WT-clone, PRVABC59 ZIKV or 100 µL of HBSS as a mock infection. Mouse weight was measured daily until endpoint and serum was collected from a retro-orbital bleed 2 days post infection (DPI) to quantify viremia during early infection via RT-qPCR. At 4, 6, and 8 DPI, a cohort of mice were euthanized with isoflurane and perfused with 20 mL PBS before brain and spleen tissues were collected. Tissues were stored in RNAlater solution (ThermoFisher) at 4 °C until processing. To evaluate the presence of ZIKV-reactive IgG, mice were euthanized with isoflurane at 20 DPI and blood was collected via cardiac puncture. Serum extracted from these blood samples was stored at −80 °C until use.

### 2.9. Infection in Pregnant Ifnar1^−/−^ Mice

Superovulation was induced in female *Ifnar1*^−*/*−^ mice aged 10–12 weeks, as previously described [15]. Dams were mated with *Ifnar1*^−*/*−^ sires for approximately 16 h and then separated (E0.5). At E6.5, dams were infected intraperitoneally with 1 × 10^4^ FFU of either X1 or WT ZIKV or 100 microL (µL) HBSS as a mock infection. Seven DPI on E13.5, mice were sacrificed using isoflurane, perfused with 20 mL PBS, and maternal brain and spleen were collected. The fetal outcome was assessed by the number of fetuses and resorptions present and the fetal head and placenta were collected to determine viral burden. All samples were stored in RNAlater solution (ThermoFisher) at 4 °C until processing.

### 2.10. Tissue Viral Load by RT-qPCR

Tissues collected from infected mice were normalized by weight and mechanically homogenized in TRIzol (ThermoFisher) using a BeadBug instrument (Benchmark Scientific). A standard TRIzol/chloroform RNA isolation protocol was used to reduce lipid contamination [16]. Total RNA was then isolated from the resulting aqueous layer using an E.Z.N.A. total RNA I kit (Omega Bio-Tek). An iScript gDNA clear cDNA synthesis kit (Bio-Rad) was used to reverse transcribe 1000 ng of RNA isolated from infected tissues. The resulting cDNA was used to quantify the amount of ZIKV genome present by qPCR.

### 2.11. Infection of Adult HuSTAT2 Mice

Male and female 5–7-week-old HuSTAT2 mice were infected intraperitoneally with 1 × 10^4^ FFU of either X1 or WT ZIKV or 100 µL HBSS as a mock infection. The weight of the infected animals was monitored daily. Mice were euthanized with isoflurane at 20 DPI and blood was collected via cardiac puncture. Serum isolated from the blood samples was used to determine the amount of ZIKV-reactive IgG present via indirect ELISA as well as viral neutralization by focus reduction neutralization test (FRNT).

### 2.12. ZIKV-Reactive IgG ELISA

Immulon 4HBX 96-well ELISA plates (ThermoFisher) were coated overnight at 4 °C with 200 ng/well of ZIKV virus particles diluted in PBS (pH 7.4). Plates were blocked with SuperBlock (ThermoFisher) at room temperature for 1.5 h then washed once with PBS-T (0.05% Tween 20). Sera from infected animals was serially diluted in PBS-T and incubated on the plate for 1.5 h at room temperature then washed 3 times with PBS-T. Horseradish peroxidase-conjugated donkey anti-mouse IgG antibody (Jackson Research Laboratories) was diluted 1:4000 and incubated on the plate for 45 min and then washed 6 times with PBS-T. TMB substrate (ThermoFisher) was added to the plate and allowed to develop for 3 min before 0.3 N H_2_SO_4_ was added to stop the reaction. ELISA absorbance was read at 450 nm for 0.1 s on a Victor X5 plate reader (PerkinElmer).

### 2.13. FRNT Measurement of Serum Neutralization

First, ten thousand Vero cells per well were seeded onto opaque Nunc-immuno 96-well plates (MilliporeSigma). Next, heat-inactivated sera from infected animals was serially diluted and then incubated with WT ZIKV at a concentration of 10 FFU per well at 37 °C for one hour. The resulting serum-treated virus was then used to infect the plated Vero cells for one hour at 37 °C. Cells were then overlaid with a 1:1 mixture of 2.5% Avicel (MilliporeSigma) and complete 2× MEM medium and incubated at 37 °C for 48 h. After incubation, the cells were fixed with 4% PFA then washed with PBS and stored at 4 °C until staining. The presence of ZIKV in infected cells was detected with a mouse anti-flavivirus 4G2 primary antibody (Millipore) and a donkey anti-mouse IgG horseradish peroxidase-conjugated secondary antibody (Jackson Research Laboratories), as previously described [14]. A nonlinear regression analysis was used to determine the plasma dilution factor at which 50% ZIKV neutralization occurs compared to an untreated control.

### 2.14. Statistical Analysis

GraphPad Prism 7 software (GraphPad Software) was used to analyze all data. Differences were considered statistically significant if *p* < 0.05. Specific statistical analysis methods are detailed in the figure legends of the experimental results.

## 3. Results

### 3.1. Development of an Infectious ZIKV X1 Mutant

As previously described, a highly conserved cytosine (nt 10415) located in the P2′ region of ZIKV xrRNA1 secondary structure is necessary for anti-exoribonuclease activity (Figure 1A) [1,2,3,4,5]. In the tertiary structure of xrRNA1, this cytosine forms several bonds that stabilize the phosphate backbone kink essential to inhibiting degradation by host exoribonucleases (Figure 1B). Replacing C10415 with a guanine is predicted to disrupt the tertiary structure of xrRNA1 without significantly altering the sequence (Figure 1C). We used site-directed mutagenesis to create an infectious cDNA clone of Puerto Rican strain PRVABC59 with a single nucleotide substitution C10415G, termed the ZIKV X1 mutant. Sanger sequencing confirmed the presence of the X1 mutation. To rescue infectious virus, Vero cells were transfected with RNA transcribed from the PRVABC59 clone with either the X1 mutation (X1) or no mutations (WT). Following transfection, we determined ZIKV genome copies present in the supernatant of transfected cells to evaluate X1 virus replication. We found that both WT and X1 transfections produced a mean of 2.3 × 10^12^ viral genome copies per mL with no significant difference between the WT (1.9 × 10^12^, ±1.6 × 10^11^ SEM) and X1 clones (2.7 × 10^12^, ±9 × 10^10^ SEM, *p* = 0.1, paired t-test, Figure 1D). Additionally, there was no significant difference in the amount of infectious WT and X1 virus as measured by a focus forming unit assay (FFU). In this assay, we found that the transfection of ZIKV produced 1.7 × 10^6^ FFU per mL (±2.9 × 10^5^ SEM) of supernatant, while X1 produced 8.7 × 10^5^ FFU per mL (±1.3 × 10^5^ SEM) (*p* = 0.1, paired t-test, Figure 1E). These results indicate that the X1 mutation did not significantly alter viral production after transfection and rescue of infectious clones; however, there was a trend toward less infectious viral particles in the clone-derived X1-mutant virus.

### 3.2. X1 Mutation Does Not Significantly Alter Viral Growth In Vitro

To elucidate the sfRNA phenotype of our X1 infectious clone, the production of sfRNAs during infection was evaluated by Northern blot of the total RNA from infected A549 cells (Figure 2A) [1,2,3]. When compared to cells infected with WT ZIKV, we determined that the X1 virus only produces sfRNA2 and lacks any observable sfRNA1 production in both biological replicates (Figure 2A). During these experiments, far less RNA was collected from WT infected cells, due to extensive cytopathic effect. Interestingly, we observed that an sfRNA3 fragment became prominent upon loss of xrRNA1. This sfRNA3 band was likely generated by additional unknown mechanisms within the infected cell, independent of the ability for xrRNA1 to resist exonucleolytic decay. Next, we inoculated human A549 cells and U4.4 cells (*Aedes albopictus* cells) with clone-derived X1 and WT viruses (Figure 2B–E). Confluent cells were infected at an MOI of 0.1 and supernatant was harvested at 0, 24, 48, and 72 h post infection (HPI) to measure the viral genome and infectious virus using RT-PCR and FFU, respectively. We found that the extracellular ZIKV genome increased with similar kinetics over 72 h following inoculation with X1 and WT clone-derived virus in A549 cells (2-way ANOVA, *p* = 0.9423, Figure 2B). Using viral titer as measured by FFU, we also found no significant difference in infectious virus titers over 72 h when comparing clone-derived X1 and WT viruses (2-way ANOVA, *p* = 0.4603, Figure 2C). However, in U4.4 cells, we observed that both X1 genome copies (2-way ANOVA, *p* = 0.9134, Figure 2D) and infectious viral titer (2-way ANOVA, *p* = 0.4782, Figure 2E) increased during infection, maintaining similar growth kinetics between time points. Percent specific infectivity of these infections was found to be 0.06% for WT and 0.05% for X1.

These data demonstrate that the clone-derived X1 ZIKV results in the loss of sfRNA1 expression with a single nucleotide substitution, and the loss of sfRNA1 production does not significantly alter viral growth in cultured A549 cells.

### 3.3. X1 ZIKV Is Attenuated in Adult Ifnar1^−/−^ Mice and Produces Neutralizing Antibody Responses

Since the WT ZIKV infectious clone is attenuated in wild-type mice [13], we compared acute infection with X1, WT ZIKV, or the original clinical isolate PRVABC59 in adult *Ifnar1*^−*/*−^ mice. This model has been established as an important and relevant model of acute ZIKV infection in mice [17]. Moreover, infection of *Ifnar1*^−*/*−^ mice was previously shown to restore the virulence of a the WNV clone, in which production of sfRNA1 and sfRNA2 was eliminated through several deletion and substitution mutations to wild-type levels [10]. *Ifnar1*^−*/*−^ mice aged 5–7 weeks were inoculated by intraperitoneal (IP) injection with 10^4^ FFU of X1, WT, or PRVABC59 (*n* = 5). To determine early viral load, we performed a retro-orbital bleed at 2 DPI and measured the amount of ZIKV genome present in the serum via RT-qPCR. We found that all mice infected with PRVABC59 or WT ZIKV had a quantifiable viral genome (10^5^–10^6^ copies/mL serum) present with no significant difference between the two infections (Figure 3A). However, only 2 of the 5 mice infected with X1 had a measurable viral genome in the serum at 2 DPI (Figure 3A). Additionally, *Ifnar1*^−*/*−^ mice inoculated with the X1 mutant virus gained weight during acute infection with an average increase of 3.8% at 6 DPI (Figure 3B). This weight gain was seen in all X1 infected mice regardless of presence of detectable virus at 2 DPI and was significantly higher than mice infected with WT ZIKV or PRVABC59, which lost an average of 3% and 1% body weight respectively at 6 DPI (2-way ANOVA, *p* < 0.05, Figure 3B). While some mice exhibited up to 6% weight loss during these studies, no mortality was observed in any experimental group.

We also assessed the antibody response elicited by these infections at 20 DPI via indirect ELISA using purified ZIKV particles as antigen and a donkey anti-mouse IgG HRP conjugate antibody to detect ZIKV-reactive IgG in infected mouse sera by ELISA. We found that despite the attenuated phenotype of X1 ZIKV, the X1 ZIKV-inoculated adult *Ifnar1*^−*/*−^ mice exhibited equivalent production of ZIKV-reactive IgG antibodies when compared to WT inoculated adult *Ifnar1*^−*/*−^ mice (Figure 3C). Together, these results show that WT ZIKV and PRVABC59 produce similar acute infections, while X1 exhibits attenuation while generating an antibody response comparable to WT ZIKV and PRVABC59 in *Ifnar1*^−*/*−^ mice.

### 3.4. X1 Is Attenuated in the Central Nervous System (CNS) Tissue of Adult Ifnar1^−/−^ Mice

Because the ZIKV X1 mutant virus exhibited attenuated acute in vivo infection despite the lack of a functional type I IFN response, we sought to determine if this attenuation was more clearly observed in different tissues. Specifically, we investigated the impact of sfRNA1 deficiency on ZIKV infection of the brain in the absence of a type I IFN response. We inoculated adult *Ifnar1*^−*/*−^ mice with 10^4^ FFU (10^7^ genome equivalents, IP inoculation) of WT or X1 (*n* = 6) then collected brain and spleen from both groups at 6 DPI. We found that X1 exhibits differential patterns of infection dependent on tissue. The spleens of *Ifnar1*^−*/*−^ mice (5–7 weeks old) infected with either WT or X1 exhibited similar ZIKV genome copies at 6 DPI (1.6 × 10^4^ genome copies/mg tissue, Mann-Whitney Test, *p* = 0.3939, Figure 4A). However, in the brain, we found that X1 exhibited a 78% decrease in genome copies per mg of tissue compared to WT (Figure 4B). This decrease was significantly different and replicated across multiple repetitions of the experiment (Mann-Whitney Test, *p* = 0.0022, Figure 4B). To further define this tissue-specific pattern of X1 ZIKV infection, we expanded the time frame of sample collection to include 4 and 8 DPI. Despite the differences in growth in the brain, we observed that both the X1 and WT genome in the spleen peaked early during infection at 4 DPI with a mean of 3.6 × 10^5^ genome copies per mg of tissue (Figure 4C). Indeed, no significant differences in viral genome in the spleen were seen between X1 ZIKV and WT ZIKV at all time points (Figure 4C). Further reinforcing our findings at 6 DPI, we found a two-fold decrease in X1 ZIKV viral genome in brain tissue compared to WT ZIKV during early (4 DPI) and later stages (8 DPI) of acute infection (**p* = 0.003, Two-way ANOVA, Figure 4D). Additionally, no mortality was observed for either infection condition. To determine that the X1 virus had not reverted and was still expressing knock-down of sfRNA1, we detected the presence of sfRNA via Northern blot of RNA from the spleen of infected animals at 2 DPI. We observed that sfRNA phenotype observed in vitro (Figure 2A) was also seen in vivo (Figure 4E). Overall, X1 ZIKV is able to replicate the infection kinetics of WT ZIKV in the spleens of *Ifnar1*^−*/*−^ mice but exhibits tissue-specific restriction in viral growth in brain tissue. These data indicate that the attenuation of X1 infection is specific to tissues and that this attenuation is not simply due to a temporal lag in X1 ZIKV infiltration of the CNS.

### 3.5. X1 ZIKV Replication Is Attenuated in the Placenta of Pregnant Ifnar1^−/−^ Mice

ZIKV vertical transmission to the placenta and fetal tissue are important complications of infection. To further define the impact of the X1 mutation on in vivo infection, we characterized X1 ZIKV infection in a pregnant mouse model. Superovulation was used to induce timed pregnancy in *Ifnar1*^−*/*−^ dams mated with *Ifnar1*^−*/*−^ sires (*n* = 5, Figure 5A). Dams were infected at embryonic day 6.5 (E6.5) with X1, the clone-derived WT as a positive control, or HBSS to serve as an uninfected control when evaluating fetal outcome via reabsorption. We assessed fetal outcome in these groups at E13.5 (7 DPI) and found that infection with X1 did not induce fetal reabsorption to the extent WT infection had (Figure 5B). ZIKV genome quantified with RT-qPCR of RNA isolated from maternal spleen (Figure 5C) and brain tissue (Figure 5D) showed patterns of X1 and WT infection similar to those seen in adult *Ifnar1*^−*/*−^ mice (Figure 4). Similarly, no mortality was observed in any experimental group. No measurable viral genome was detected in the heads of fetal mice from either infection group at this time point (Figure 5E). However, we found that both WT and X1 infected the placenta, and WT-infected placenta exhibited significantly higher ZIKV genome copied, 10^5^ copies/mg tissue, than X1 infected placental tissue, 10^3^ copies/mg tissue (*p* < 0.01 via Mann-Whitney test, Figure 5F). These findings show for the first time that the targeted mutation of xrRNA1 to eliminate sfRNA1 production results in tissue-specific attenuation of ZIKV replication in the placenta.

### 3.6. Attenuated X1 ZIKV Generates a Strong Neutralizing Antibody Response in Transgenic STAT2 Mice

Use of the *Ifnar1*^−*/*−^ mouse model has provided us with important insights into the pathogenesis of our X1 mutant ZIKV, but this is an immunocompromised model which limits the interpretation of attenuation, due to the loss of type I IFN stimulation. The recently established mouse model with a knocked-in human STAT2 gene (h*STAT2* KI) provided an opportunity to assess X1 pathogenesis in an immunocompetent animal model [18]. Adult (5–7-weeks-old) h*STAT2* KI mice were infected via IP with 10^4^ FFU of either X1, WT ZIKV, or 100 µL of HBSS as a mock infection control (*n* = 6). There was no detectable viral genome in the serum of infected animals at 2 DPI and no weight loss was observed during acute infection with either WT or X1 ZIKV (Figure 6A). At 20 DPI, we found that ZIKV-reactive IgG antibodies were detectable in the serum of both WT ZIKV and X1 infected mice (Figure 6B). With an average OD450 nm measurement of 0.56, ZIKV X1 infected mice exhibited significantly higher levels of IgG than those infected with WT ZIKV (OD450 nm 0.38, 2-way ANOVA, *p* ≤ 0.002, Figure 6B). To determine if serum from infected animals could neutralize WT ZIKV in comparison to mock infected animals, we performed a focus-forming unit reduction neutralization test (FRNT). Serum from mock infected mice did not neutralize or reduce the infectivity of WT ZIKV virions (Figure 6C). The serum from all animals infected with either X1 or WT ZIKV was able to reduce ZIKV infectivity with Log_10_ EC50 values at −2.98 and −2.88 respectively (Figure 6C). These data illustrate that, despite the attenuated phenotype of the X1 mutant in *Ifnar1*^−*/*−^ mice, the X1 virus produced a strong neutralizing antibody response comparable to WT ZIKV in the h*STAT2* KI model of ZIKV infection.

## 4. Discussion

We utilized the ZIKV xrRNA1 tertiary structure to construct a mutant with a single nucleotide change that disrupts xrRNA1 structure and eliminates sfRNA1 production without significantly altering the sequence of the ZIKV 3′ UTR. We show for the first time that the targeted mutation of xrRNA1 results in the loss of sfRNA1 production and tissue-specific attenuation of ZIKV replication in brain tissue and placenta tissue of type I interferon receptor knockout mice. Despite the attenuated phenotype of X1 ZIKV, type I interferon receptor knockout mice and transgenic h*STAT2* KI mice develop robust antibody responses. This is likely due to the unchanged replication of the X1 mutant virus in the spleen of *Ifnar1*^−*/*−^ mice compared to the wild-type virus and supports the continued study of xrRNA-dependent attenuation strategies as potential attenuated vaccine approaches in the future.

Our work has several strengths that add to our current understanding of the role of xrRNA1-dependent production of sfRNA. Previous studies have shown that sfRNA biogenesis is necessary to limit RNA1 and Toll pathway activation in arthropods, to limit mRNA decay in host cell cytoplasm, and to impede antiviral type I IFN responses in mammalian cells [4,5,6,7,8,9,10,11,12,13,14,15,16,17,18,19,20,21,22]. These previous studies utilized the extensive sequence and structural mutation of the 3′ UTR or the total elimination of sfRNA1 and sfRNA2 production. Our approach utilized information of the xrRNA1 structure to make mutations that disrupt tertiary structure without significantly altering the sequence of the 3′ UTR. Thus, our findings are directly related to the structure–function relationships of xrRNA1 and sfRNA1 production.

Additionally, the scope of the current published literature includes little information on the role of xrRNA1-dependent sfRNA1 production in the pathogenesis of flavivirus infections in animal models. Early investigations with sfRNA1-deficient WNV found evidence of decreased cytopathicity in cell culture and decreased mortality in the mouse model [2]. Here, we show for the first time that ZIKV xrRNA1 mutation results in tissue-specific attenuation of viral replication in the brain and placenta in type I IFN receptor knockout mice. Previous studies have also shown that type II and type III IFN responses are critical to the control of acute RNA virus infections in the brain and placenta, respectively [23,24,25,26]. Recent studies investigating the antiviral response to alphavirus infection of the central nervous system (CNS) have identified IFN-γ and the type II IFN response as being vital for the control of RNA viruses [23,24,25,26,27]. Given that we found differences in type I IFN deficient mice, future studies evaluating the interactions between xrRNA1 mutant ZIKV and type II and III IFN responses will be critical to fully understand the mechanisms of the sfRNA1 function in different tissues.

We also found a novel infection phenotype in pregnant *Ifnar1*^−*/*−^ mice. Numerous mouse pregnancy infection models have been established to investigate the pathogenesis of congenital Zika virus syndrome as extensively reviewed by Caine et al. [28]. However, there have been no findings to date studying sfRNA biogenesis and the impact on pathogenesis in murine pregnancy models of ZIKV. Our studies have identified that the sfRNA1-deficient ZIKV X1 mutant does not efficiently infect the placenta and exhibited decreased fetal reabsorption compared to WT ZIKV infection. Since we did not detect virus in fetal brain tissue, these findings also imply that placental infection plays an important role in ZIKV-induced fetal injury.

The lack of detectable clone-derived WT ZIKV or X1 mutant in the fetal head of infected fetuses identifies an important limitation of this model and corroborates findings that infection with mouse-adapted ZIKV strains produces a more consistent fetal infection than common lab strains [29]. Future studies using mouse-adapted ZIKV or the insertion of envelope protein virulence determinants, as recently described, along with our identified xrRNA1 mutations would likely provide more information on the pathogenesis of ZIKV with and without xrRNA1 mutations in fetal tissues [29].

The 3′ UTR is a common target for attenuation approaches and vaccine development strategies for flaviviruses, although the mechanism for this attenuation is often not well defined [30,31,32,33,34]. Here, we have found that ZIKV containing a targeted mutation in the xrRNA1 structure induces a strong ZIKV-specific antibody response in both immune-deficient and immunocompetent mouse models despite the attenuated phenotype. This is especially intriguing given that the xrRNA1 structure altered in our studies is also present in other flaviviruses of global health importance like DENV and WNV [1,2,3,4,5,6,7,8,9,10,11,12,13,14,15,16,17,18,19,20,21,22,23,24,25,26,27,28,29,30,31,32,33,34,35]. Future studies targeting specific mutations to the xrRNA1 structure in other flaviviruses may provide a novel attenuation approach for vaccine development in ZIKV as well as other flaviviruses.

Previous studies have shown that sfRNA1 deficiency does not alter the growth of WNV or DENV in cultured host cells but does result in reduced cytopathic effect and plaque size [6,7,8,9,10,11,12,13,14,15,16,17,18,19,20,21,22,23,24,25,26,27,28,29,30,31,32,33,34,35,36]. Differential expression of sfRNA isotypes assists in host adaptation for some flaviviruses [35,36,37,38]. However, ZIKV consistently produces equal amounts of sfRNA1 and 2 in both mosquito and mammalian host cells, suggesting that the production of a specific sfRNA species is not crucial for survival in different hosts [1,2,3,4,5,6,7,8,9,10,11,12,13,14,15,16,17,18,19,20,21,22,23,24,25,26,27,28,29,30,31,32,33,34,35,36,37,38]. We show that xrRNA1 mutant ZIKV maintains a loss of sfRNA1 expression in splenic tissue following infection. This is the first study to show sfRNA patterns of ZIKV in the murine model of disease. We also show that sfRNA1-deficient ZIKV X1 virus displayed reduced ability to replicate in the brain and placenta. Due to low genome copy numbers in our model, we believe that infectious virus particles in these tissues are likely below the limit of detection for our FFU assay. Future studies will need to evaluate the effects of sfRNA production on infectious titers in murine models that support higher viral loads in the target tissues. We believe our study provides an important starting point for future studies that investigate the tissue-specific role of sfRNA1 and sfRNA2 production during flavivirus infection.

In conclusion, we have developed and characterized an sfRNA1-deficient ZIKV virus through minimal sequence manipulation of a 3′ UTR RNA structure and this approach can be utilized to expand the understanding of sfRNA function in ZIKV and related flaviviruses. The validation of this model for disrupting sfRNA production can also be applied to ZIKV xrRNA2 or other xrRNA structures for other vector-borne flaviviruses. Further studies examining the role of sfRNA1 and sfRNA2 expression in a tissue-specific manner will provide novel insight into the pathogenesis of flavivirus infections and may provide novel attenuation approaches for vaccine development.

## Figures and Tables

**Figure 1 viruses-12-01177-f001:**
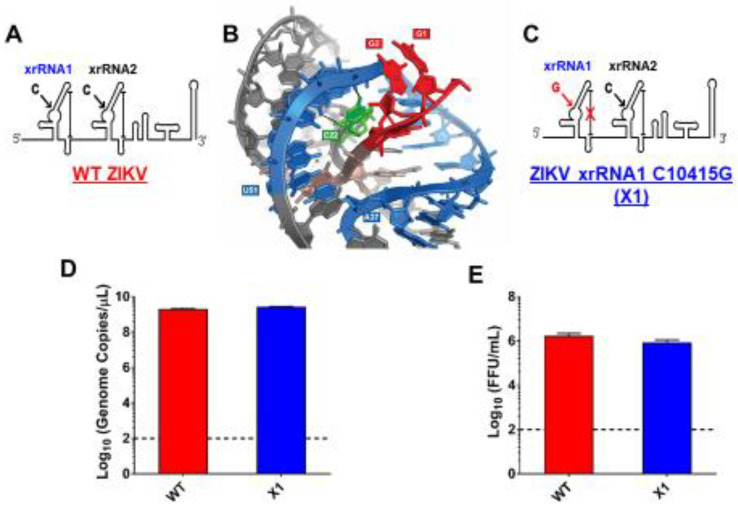
Development of infectious Zika virus (ZIKV) with discreet xrRNA1 structural mutation. (**A**) The 3′ UTR of ZIKV encodes two host exoribonuclease resistant RNA structures (xrRNAs). (**B**) The tertiary structure of ZIKV xrRNA1 has been previously described, revealing that a single cytosine at position 10,415 (green) of the crystalized RNA fragment is necessary for stabilizing the phosphate backbone kink (blue) which protects the viral 3′ UTR (red) from exoribonuclease degradation. We produced a ZIKV mutant (called X1) in which the cytosine at position 10,415 of xrRNA1 has been replaced with a guanine (**C**), weakening the tertiary structure of the RNA. A wild type (WT) ZIKV clone with no mutations (**A**) was also produced alongside the X1 mutant as a positive control. The viral genomes of either X1 or the WT clone were transfected into Vero cells. At 12 days post transfection, the amounts of both viral genome (**D**) and infectious virus (**E**) rescued from the X1 transfection were comparable to those rescued from cells transfected with WT ZIKV RNA (*n* = 6).

**Figure 2 viruses-12-01177-f002:**
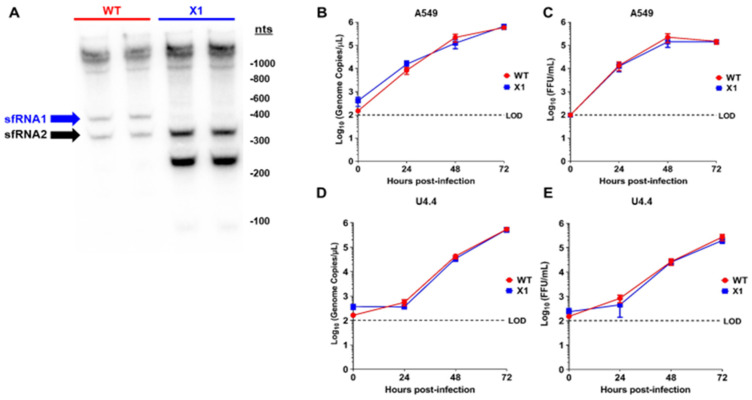
In vitro sfRNA production and viral growth kinetics of ZIKV X1 compared to WT ZIKV clone. (**A**) A549 cells were infected with either X1 or WT clone at a MOI of 1. At 48 h post infection (HPI), cellular RNA was collected and the presence of ZIKV sfRNAs was detected in two biological replicates per infection via Northern blot using a ZIKV 3′ UTR-specific probe. To analyze viral growth kinetics, human A549 cells (**B**), (**C**) or Aedes albopictus U4.4 cells (**D**), (**E**) were infected with X1 or WT at an MOI of 0.1. At 0, 24, 48, and 72 HPI, supernatant was collected and used to measure either extracellular viral RNA via RT-qPCR (**B**–**D**) or infectious virus via FFU assay (**C**–**E**). (**B**–**E**) Dashed lines represent the limit of detection (LOD). Error bars indicate standard error of the mean for six replicates across two independent experiments. (*n* = 6, NS by two-way ANOVA).

**Figure 3 viruses-12-01177-f003:**
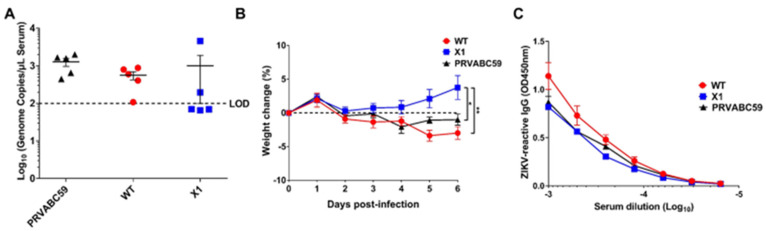
X1 infection compared to WT Clone and ZIKV PRVABC59 in adult *Ifnar1*^−*/*−^ mice. Male and female *Ifnar1*^−*/*−^ mice 5–7 weeks in age were infected via intraperitoneal (IP) injection with 1e4 FFU of X1, WT ZIKV Clone, or PRVABC59 virus. (**A**) Serum samples were collected via retro-orbital (RO )bleed at 2 days post infection (DPI) to quantify early viral infection. RNA was isolated from the sera and used to detect ZIKV genome via RT-qPCR. Dotted line represents the limit of detection (LOD), no significant differences were found using Mann-Whitney tests to compare between viral infections (*n* = 5). (**B**) The weight of infected mice was monitored during acute infection and shown here as the percent of weight change relative to the baseline set at 0 DPI. Dotted line symbolizes 0% weight change. Two-way ANOVA was used to make multiple comparisons at each time point, asterisks representative of the following: * X1 vs. PRVABC59: *p* = 0.0464 at 5 DPI and *p* = 0.0014 at 6 DPI. ** X1 vs. WT: *p* < 0.0001 at 5 and 6 DPI. (*n* = 5). (**C**) Serum was collected via cardiac stick at 20 DPI to detect a ZIKV-reactive antibody response. ZIKV-reactive IgG was detected by indirect ELISA using ZIKV virions as antigen and donkey α mouse IgG HRP conjugate as detecting antibody. Data shown as optical density (OD) at various serum dilutions, normalized with uninfected mouse sera (*n* = 2, NS).

**Figure 4 viruses-12-01177-f004:**
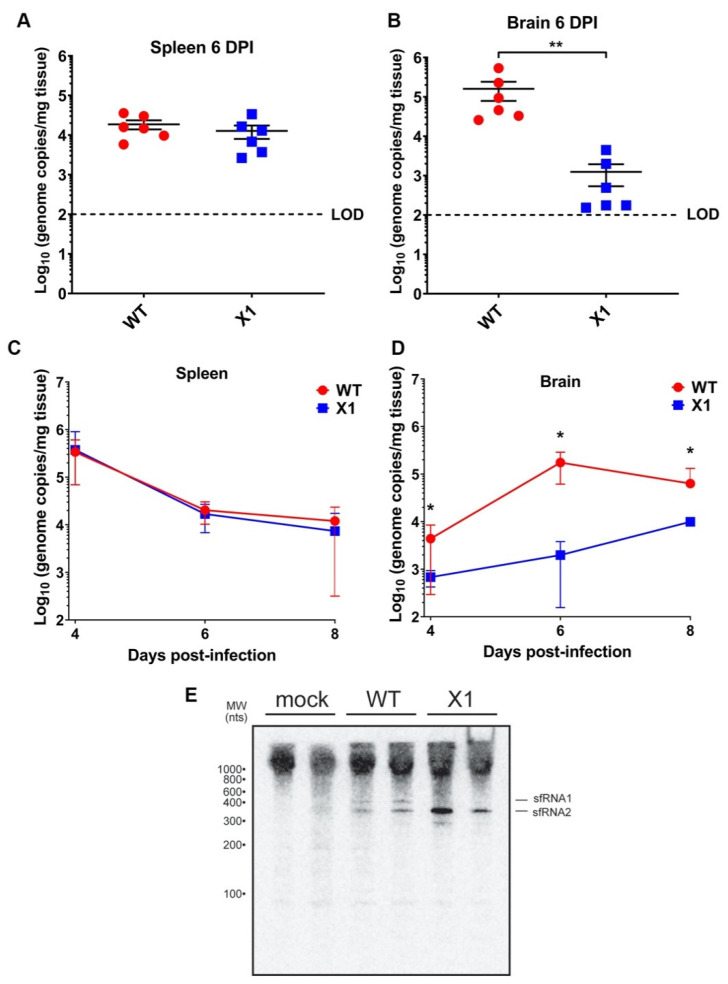
X1 ZIKV exhibits tissue-specific attenuated viral replication in *Ifnar1*^−*/*−^ mice. Male and female mice were infected with 1e4 FFU (IP) of either WT or X1 virus. At the indicated days post-infection, mice were sacrificed and perfused with PBS before harvesting the spleen (**A**–**C**) and brain tissue (**B**–**D**). Total RNA was isolated from the tissue and used to quantify ZIKV genome via RT-qPCR. Dotted line represents the limit of detection (LOD). The presence of sfRNA in infected spleens from 2 DPI was determined via northern blot (**E**). Error bars indicate the standard error of the mean for six replicates across two independent experiments. [*n* = 6 per group, ** *p* < 0.001 by Mann-Whitney test (**B**) and two-way ANOVA, * *p* < 0.01 (**D**)].

**Figure 5 viruses-12-01177-f005:**
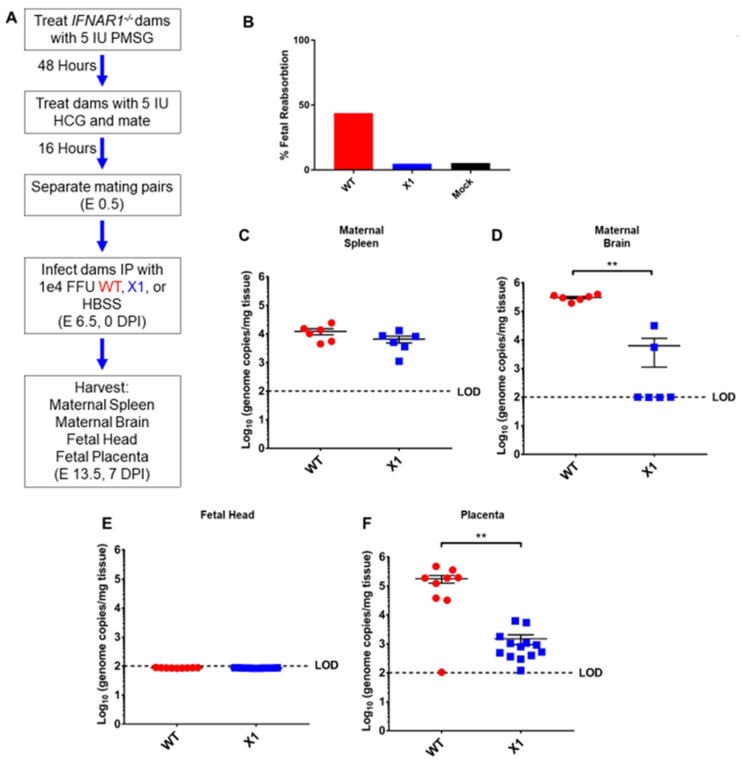
Fetal outcome and viral burden in pregnant *Ifnar1*^−*/*−^ mice infected with either WT or X1 ZIKV. (**A**) Superovulation was induced in 10–12-week-old *Ifnar1*^−*/*−^ dams which were then mated with *Ifnar1*^−*/*−^ sires. Dams were then infected IP with either 1e4 FFU of WT ZIKV clone, the X1 mutant, or 100 uL HBSS at E 6.5. At E 13.5, 7 DPI dams were sacrificed and perfused with PBS. Fetal outcome was assessed and is shown % fetal reabsorption (**B**). Maternal spleen (**C**) and brain (**D**) tissues were collected to assess viral burden in these tissues by RT-qPCR of ZIKV genome (*n* = 6, ** *p* < 0.01 via Mann-Whitney Test). The fetal head (**E**) and placenta (**F**) were also collected to assess viral burden via detection of ZIKV genome (WT *n* = 9, X1 *n* = 13, ** *p* < 0.01 via Mann-Whitney Test). Fetal data representative of 9–13 fetuses from three pregnant dams for each infection. The dotted line shows the limit of detection (LOD) for the assay, error bars indicate the standard error of the mean for these experiments. WT virus in red, ZIKV X1 virus in blue.

**Figure 6 viruses-12-01177-f006:**
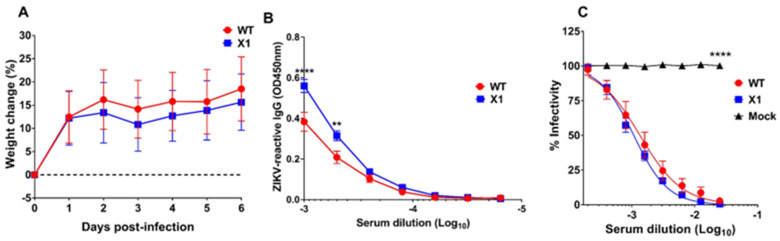
Outcomes of X1 infection compared to WT in h*STAT2* KI mice. Male and female h*STAT2* KI mice 5–7 weeks of age were infected IP with 1x10^4^ FFU of X1, WT ZIKV Clone, or HBSS as a mock infection. (**A**) Mouse weight was monitored during acute infection and shown here as the percent of weight change relative to the baseline set at 0 DPI. The dotted line symbolizes 0% weight change with no significant differences detected via two-way ANOVA. (**B**) Serum was collected via cardiac stick at 20 DPI to detect a ZIKV-reactive antibody response. ZIKV-reactive IgG was detected by indirect ELISA using ZIKV virions as antigen and donkey α mouse IgG HRP conjugate as detecting antibody. Data shown as optical density (OD) at various serum dilutions, normalized with mock infected mouse sera. (**C**) Serum from 20 DPI was also used to detect neutralization of ZIKV via FRNT. Results are shown here as the change in percent infectivity of ZIKV compared to untreated ZIKV. All data are representative of six replicates across two independent experiments, error bars indicate the standard error of the mean. (*n* = 6, **** *p* < 0.0001, ** *p* = 0.002 via two-way ANOVA).

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
