# Peer review of "Disruption of Zika Virus xrRNA1-Dependent sfRNA1 Production Results in Tissue-Specific Attenuated Viral Replication"

_viruses, 2020, doi:10.3390/v12101177_

Round 1

Reviewer 1 Report

The manuscript by Sparks et al describes the role of sfRNA in Zika virus infection in whole animal models. They found that the mutation that leads to the absence of sfRNA1 leads to reduced genome replication in the brain and placenta while not affecting the viral replication in the spleen. The authors present evidence for tissue-specific attenuation of the virus. Despite the attenuation, the mutant virus developed a robust neutralizing antibody response. Throughout the manuscript, the authors use different RNA concentrations (per ml, per ml, per mg, etc). Titer FFU per ml and genome copies per µl are used to measure viral load. This causes unintentional confusion and has to be addressed. As the viral genome copies recovered from tissues are very low, the authors need to show that the genome copies retrieved from the brain and placenta have infectious viruses and show the difference in virus titer between WT and mutant virus.

Comments
Change the genome copies figure 2 from µl to ml. It is good to be consistent to show data in genome copies per ml and virus titer per ml. As represented, the graphs appear to have similar Y values and can be misleading.
Figure 1: What is the reduction in specific infectivity? It looks like there are more RNA and less titer in the mutant. It will become more prominent is the linear range or in percentage reduction. It will be interesting to know how packaging is affected in the mutant virus and whether more noninfectious RNAs are packaged.
Have the authors tested for infectious virus (FFU) collected from animal tissues? The ‘viral load’ presented in RNA molecules appears to be low, considering the specific infectivity of flaviviruses is in the 10^3 range. Specifically, if there are 10^3 copies per mg of tissue, is there any infectious virus present in these tissues?

Minor comments
Abstract: line 13 – explain derived?
Reformat: superscripts and subscripts throughout the manuscript
210: “would be predicted”
217: Indicate the error value here for both WT and mutant RNA. Specifically, the values may become more significant when expressed per ml.
217-220: Represent the titer also in a similar way, indicate the standard error. It is considered significant when there is a 48% reduction in virus titer, as shown here.
313: To be consistent, please indicate 104 FFU is equivalent to how many RNA genome copies are used for infection.
318: A 78% reduction is described as significant here.
475: is it the correct conclusion from this study?
448: Why is antibody response by X1 strain surprising?
How stable is the mutation? Was the X1 virus sequenced after infection? Any revertants screened?

Reviewer 2 Report

Zika virus belonging to genus flavivirus has been recognized as a public health threat because it causes microcephaly in baby in addition to encephalitis like other flaviviruses. In the flavivirus infected cells, sfRNAs which are derived from 3’UTR of flavivirus genome, are accumulated and are known to play some roles in inhibition of viral RNA genome and control of immune response in host cells. In the prepared manuscript, authors showed that ZIka virus lacking xrRNA1 exhibited attenuated phenotype in mice and lower replication of this virus was detected in brain but not in spleen. Thus production of sfRNAs may relate to tissue specificity and disruption of xrRNA forming sequence by single mutation could be alternative strategy to develop attenuated vaccine. The manuscript is well organized but several points were raised for revision.

Major comments

  1. Do authors check differences in sfRNA production in various tissues (brain vs spleen)? This is important to conclude that disruption of xrRNA1 is related to attenuation.
  2. 2A; Two lanes of WT and X1 samples were run. What the differences of lane1 and 2, 3 and 4?
  3. 2A; Have authors tested similar experiment by using U4.4 cells?
  4. 2A; Can we conclude that X1 virus could produce larger amount than WT virus in A549 cells?
  5. Line 284-285; Please show mortality rates of each group.
  6. 3B; If the data obtained from mice that not detected viral genome at 2 DPI are eliminated, was the similar curve observed with WT?

Minor comments

  1. Line 89; Please put an appropriate reference.
  2. Line 123 and others; 7 should be superscripted. Please check throughout the manuscript carefully.
  3. Line 240; fig 2A should be Figure 2A.
  4. FRNT and DPI were defined several times. Please check carefully.

Round 2

Reviewer 2 Report

Authors revised the manuscript accordingly, thus the revised manuscript is now considered to be published in "Viruses".